# Violence and Clinical Learning Environments in Medical Residencies

**DOI:** 10.3390/ijerph20186754

**Published:** 2023-09-13

**Authors:** Liz Hamui-Sutton, Francisco Paz-Rodriguez, Alejandra Sánchez-Guzmán, Tania Vives-Varela, Teresa Corona

**Affiliations:** 1Division of Graduate Studies, Faculty of Medicine, National Autonomous University of Mexico, Unidad de Posgrado, Coyoacán, Mexico City 04510, Mexico or lizhamui@comunidad.unam.mx (L.H.-S.); alesanchezguz@comunidad.unam.mx (A.S.-G.); coronav@unam.mx (T.C.); 2Laboratory of Clinical Neuropsychology, National Institute of Neurology and Neurosurgery, Av. Insurgentes Sur 3877, La Fama, Tlalpan, Mexico City 14269, Mexico; 3Department of Research in Medical Education, Faculty of Medicine, National Autonomous University of Mexico, Av. Universidad 3000, Coyoacán, Mexico City 04510, Mexico; tania.vives@comunidad.unam.mx; 4Clinical Laboratory of Neurodegenerative Diseases, National Institute of Neurology and Neurosurgery, Av. Insurgentes Sur 3877, La Fama, Tlalpan, Mexico City 14269, Mexico

**Keywords:** violence, gender identity, clinical settings, medical residencies, inappropriate behaviors

## Abstract

Introduction: The objective of this study was to describe and analyze residents’ perceptions of characteristics on the expansive/restrictive continuum of their clinical learning environment. Methods: We conducted a quantitative, descriptive and cross-sectional study. A self-administered questionnaire was designed, programmed and applied to residents at the Faculty of Medicine of the National Autonomous University of Mexico. The instrument was structured in eight sections, and for this article, Section 3, which referred to clinical environments and violence was considered. The questionnaire had an 85% response rate, with 12,612 residents from 113 medical units and 78 specialties participating. The reliability and internal consistency measured with alpha omega obtained a value of ω 0.835 (CI; 0.828–0.843). Results: Unpleasant, competitive, tense and conflictive contexts were related to restrictive environments. Sexual orientation influenced the perception of intolerance in the clinical setting with respect to discriminatory comments, such that for gender minorities, the environment was experienced as exclusionary. First-year residents perceived environments as more aggressive, a perception that tended to decrease in later years of residency. Discussion: Abuses in power relations, rigid hierarchical positions and offensive clinical interactions may foster restrictive environments. In such settings, the reproduction of socio-culturally learned violence is feasible; however, asymmetrical relationships may be deconstructed and transformed.

## 1. Introduction

Violence in institutional contexts is a social problem that is reproduced and manifested in various forms [1,2], including psychological, physical, sexual, workplace and even cyber abuse. Abuse is understood as the excessive, unreasonable, inappropriate and unjust misappropriation of authority, which harms others and whose expression may range from minor details of disparagement to severe acts of aggression that constitute crimes [3]. In this article, the term violence is used to describe a wide range of aggressions, while the concept of abuse refers to asymmetrical power relations in which an individual or group harms another. Abuse is limited to situations in specific environments and constitutes just one of the multiple manifestations of violence [4]. The social sphere of the clinic is an organization constructed through interactions among health professionals, patients and their families, where residents demonstrate their knowledge and medical reasoning, make decisions and put procedures and techniques into practice [5,6].

The contributions of this study focus on the existing link between clinical environments and expressions of violence from the resident’s perspective. The breadth of the participants in this study, in addition to the diversity of specialties and medical units, offers the possibility of a detailed analysis regarding factors related to violence expressions at the hospital. For example, it is of interest to address the perception of intolerance or discrimination according to gender identity and sexual orientation, as well as expressions of violence by year of residence, which has been little studied in the literature. The hypothetical questions posed in the questionnaire seek to fill the scientific gap to understand the relationship between organizational culture, social interactions and perceptions among residents that may reproduce various types of abuse.

Organizations are systems that seek to coordinate the activities of two or more individuals to achieve common goals, requiring constant cooperation and communication. Structurally, hospitals constitute a realm where differences are formally established according to institutional norms and regulations, as well as the distribution of positions [7]. However, informal power relations also coexist among the agents. In day-to-day activities, inequalities based on work, gender, academic level, class, religion and others [8] are recreated in social interactions. Different interests converge in the healthcare process, and violence and the breakdown of communication may emerge [9].

The five basic premises on violence used in our study are as follows: (1) violence cannot be reduced to individuals or society alone; (2) violence performs social functions of differentiation and integration in the social order; (3) violence creates relational processes of (dis)articulation of meaning; (4) violence is socially defined in accordance with legal and moral criteria; and (5) societies survive violence through conflict [10]. In clinical settings, expressions of different types of violence are frequent, involve individuals and collectivities and disrupt the logic of cooperation and organizational communication in healthcare processes [11,12]. Abuse, as defined above, constitutes a form of violence based on power asymmetries made visible in everyday situations.

For abuse to exist, several elements are called into play, ranging from the personal attributes of the recipient and the perpetrator to the characteristics of the work and organizational factors that affect resident training [13]. The focus on social interactions refers to the relationship between people and the environment [14]. In this study, we explored aspects related to work and organizational culture through a survey to explain the context in which violence emerges.

In his social learning theory, Bandura [15] considers violence as a behavior learned by observation or experience. However, observing aggressive models is not enough to exercise violence; the sociocultural environment plays a decisive role in its execution. Violent behavior can be controlled by its consequences; therefore, it may be modified when the effects produced are altered. The modeling of non-violent behaviors in clinical learning environments can have a positive influence in limiting violence. For example, cooperative interactions promote attitudes of sympathy, solidarity and reciprocity [16].

Learning in clinical contexts occurs at the same time as activities related to health care. Hospital environments may be learning-oriented to a greater or lesser degree and are located on a continuum that ranges from expansive to restrictive [17,18]. Expansive learning environments are conducive to the inclusion of apprentices in the tasks of the department to which they belong, promote their participation with explicit institutional recognition manifested by structural support, favor constant supervision of clinical practice, establish positive relations between professors and residents, motivate the search for knowledge and encourage research and teamwork [19]. In contrast, restrictive environments are characterized by narrow access to resident learning, ambivalence toward their position in the department, lack of personal support, limited opportunities for participation and intimidating and even violent environments [20]. For residents, it is important to adjust to the departmental culture, understand the preferences and idiosyncrasies of others and learn how to work as a team without having to engage in constant negotiations [21]. When residents encounter barriers in social practices and are excluded from the team, learning may be impacted [22,23]. Different types of violence often emerge in restrictive environments in which power relations are more pronounced and communication less fluid.

Workers in the health sector (undergraduate and graduate-level physicians, attending physicians, specialists and nurses, among others) are at risk of suffering acts of interpersonal violence in their workplaces perpetuated by superiors, colleagues and even by patients, with women often being the primary targets [24]. As Stone et al. [25] explain, medicine is an immersive career; clinical work is characterized by structured breakdowns in barriers to intimacy, and residential requirements such as night rotations can blur the lines between work and social life.

Physicians abused by other physicians constitute an invisible population of victims, prevalent in organizations that oversee professional development, such as teaching hospitals [26]. The social discourse around abuse often invalidates victims’ experiences, encourages self-blame and minimizes the situation, perpetuating violence [25]. An individualistic perspective obscures the environmental factors that influence the phenomenon of violence. The internalization of interpersonal violence by their superiors that residents often experience and observe lays the foundations for the generation and reproduction of similar behaviors toward their peers [27,28,29,30], leading to restrictive learning environments and abuse in relationships among health professionals.

The research question guiding this study was enunciated as follows: how do expressions of violence in clinical contexts alter learning environments, impacting the professional training of residents?

The objective was to describe and analyze residents’ perceptions of characteristics on the expansive/restrictive continuum of the clinical learning environment where they work and study.

## 2. Material and Methods

The specific objectives of this study were as follows:To identify differences in the perception of the environment by specialty group.To analyze the violence observed and its influence on the perception of an expansive or restrictive learning environment.To compare how men, women and gender minorities perceive the clinical learning environment.To describe the perception of institutional risk in hypothetical situations of violence based on the medical unit size by number of residents.To analyze residents’ perceptions of common expressions of violence in their workplace based on their academic year.

This is a quantitative, descriptive and cross-sectional study. The research was planned and implemented from the Postgraduate Studies Division (DEP) at the National Autonomous University of Mexico (UNAM) Faculty of Medicine. From June to December 2021, the literature was reviewed, and a self-administered questionnaire was designed taking into account different violence scales such as the study of sexual and gender harassment conducted at the Complutense University of Madrid [31], as well as the structure and operation of medical residences in Mexican health institutions. The questionnaire was reviewed and tested with social service students to validate its content and understanding of the questions, and adjustments were made accordingly. The instrument was digitally programmed at the Computer Department.

Participants in the study were the registered residents in the 2021–2022 cycle who presented the annual exam of the Academic Program for Medical Specialties (Plan Único de Especializaciones Médicas, PUEM) of (UNAM) Faculty of Medicine. When they accessed the website to consult their evaluations, they were asked to complete the survey. The inclusion criteria considered the 12,900 registered residents at PUEM, so it was more of a census. In total, 6827 were women (52.92%) and 6073 were men (47.08%), distributed in 78 medical specialties, 156 medical units and 686 programs. Residents from other universities’ academic programs were excluded. The questionnaire was available in January 2022. They answered the survey voluntarily after approving the informed consent at the beginning of the instrument. The responses were received in real time via the digital database platform. At the end of the reception period, data was reviewed to correct errors. Once clear, statistical analyses were carried out according to the objectives of the study.

The “Medical Residencies Free of Violence” (Residencias Médicas Libres de Violencia, RMLV) project was reviewed by the Research and Ethics Committees of the Research Division of the UNAM School of Medicine and registered under code FM/DI/011/2022. Prior to answering the questionnaire, residents were asked for their voluntary participation, informed consent and permission to use the data anonymously for educational and research purposes.

The structure of the RMLV questionnaire consisted of an introduction and eight sections. The introduction contained preloaded data obtained from the DEP databases: UNAM student account number (resident ID), year of residency, specialty and subspecialty, institution and medical unit. This section also included a presentation of the study objective, a checkbox confirming their voluntary participation and a hyperlink with the full informed consent document. The first and second sections (one question each) explored gender identity (how they self-identify) and sexual orientation (to whom they are attracted) in multiple-choice questions. The third section (17 questions) referred to the environment at the clinical unit and its description through qualifying adjectives and hypothetical situations related to expressions of violence, abuse and harassment. The fourth (28 questions), fifth (four question) and sixth (five questions) dealt with types of violence, gender discrimination and cyber-aggression. These questions were structured in a matrix and constituted the core of the questionnaire. The possibility of having experienced violence was explored, and if affirmative, the frequency, severity, location where it took place, time of the event, the perpetrator of the abuse, the actions carried out, reporting mechanisms and institutional response were explored with multiple-choice questions. The seventh section (four questions) alluded to the effects of violence on the health of residents, on their state of mind, the support available to address their mental health and the impact of violence on continuing the residency. Finally, the eighth section (eight questions) collected sociodemographic data of the residents. The scoring systems were diverse in each section depending on the question; sometimes yes or no, Likert scales and sentence selection were also used. The questionnaire in Spanish may be accessed at the following link: https://www.dropbox.com/s/8nqoc5sddzubvob/Cuestionario%20RMLV%20para%20protocolo%2028-01-2022.pdf?dl=0 (accessed on 10 September 2023).

Section 3 considered four sets of questions on clinical:Prompt 3.1 was comprised of items in which the residents described their workplace environment using qualifying adjectives. The dichotomous adjectives considered were unpleasant/pleasant, competitive/cooperative, tense/relaxed and conflictive/constructive, using a five-point response scale. The reliability and internal consistency obtained was 0.846 (CI; 0.841–0.850).Prompt 3.2 was phrased as follows: “What do you think would be the most typical reaction in your direct and daily environment on site (peers, colleagues) to a comment of a discriminatory nature?” There were three response options: most would disapprove, most would tolerate it and most would approve and continue with the comment.Prompt 3.3 presented five hypothetical situations of abuse and institutional consequences. The response options were dichotomous: yes or no. The reliability and internal consistency obtained was 0.774 (CI; 0.768–0.780).Prompt 3.4 consisted of seven statements related to common inappropriate practices in medical residencies with five response options ranging from strongly agree to strongly disagree. The reliability and internal consistency obtained was 0.726 (CI; 0.719–0.734).

The questions in English are presented like Appendix A as they appear in Section 3 of the RMLV questionnaire by the title “Environment at the Clinical Site”.

### 2.1. Statistical Analysis

The statistical package SPSS v.23.0 (IBM Corp. Armonk, New York, NY, USA) was used to analyze the data obtained. Descriptive statistics was used in the analysis and bivariate comparisons were made with Chi-squared and *t*-tests. Contingency tables have a moderating effect on the variables of interest, such as the type of violence (sexual, physical, psychological, academic, labor and gender), medical specialty and perception of risk of violence and their relationship with clinical learning environments. For this work, we did not seek to develop an explanatory model due to the possible number of confounding variables to be considered in the study variable (clinical setting) and their impact on the accuracy of the model. Comparisons between proportions were performed using the z-test and Bonferroni correction. A α = 0.05 was considered statistically significant.

For this reason, using contingency tables, the moderating effect of variables of interest, such as the type of violence (sexual, physical, psychological, academic, labor and gender), the medical specialty and the perception of risk of violence, could be verified.

For this article, we considered Section 3 of the RMLV questionnaire to describe violence in clinical learning environments and to situate them on the expansive–restrictive continuum. The reliability and internal consistency of violence in medical residencies were assessed with alpha omega, an internal consistency estimator that is more sensitive than other estimators (like Coefficient α, β, H Ordinal) and based on factor loadings that indicate the proportion of the variance attributed to the common variance and does not overestimate reliability [32]. The value obtained was 0.835 (CI; 0.828–0.843).

### 2.2. Conceptual Framework

Figure 1 illustrates the conceptual framework with intermediate categories for a multidimensional study analyzing violent situations. At the center appears the phenomenon of abuse, understood as a relational problem that involves individualistic and social perspectives. In the first, the psychological traits of the agents are described, focusing on the personal characteristics of the recipient of the aggression, as well as the perpetrator. From a social perspective, the subjects involved in the abusive environment are co-participants in the events. Abuse may be explicit or covert, with the latter expressed as microaggressions, i.e., multiple and recurrent subtle forms of disparagement, aggression and harassment that require the resilience of the recipient, usually expressed as resistance (mutual protection, emotional support, silence, not reporting), denunciation or inaction. Our study focused on the social perspective and symbolic violence, emphasizing the workplace and educational environment experienced by residents in the clinical setting.

The diagram is divided into two levels: institutional and interpersonal. The first refers to the exercise of clinical activity in the medical unit (the clinic understood as a social order whose goals involve the patient care process). The hospital structure is complex and constitutes a specific institutional culture predicated by a vision, mission, values and traditions that guide its operation. Ideally, organizations seek to maintain a balance between their agents, fostering fluid cooperation and communication to achieve common goals. The institutional order is supported by laws, regulations, manuals, operational programs, protocols, algorithms and documentary records that constitute the shared references framing interactions. Medical units also have elaborate administrative designs, reflected in the organization chart in which positions are assigned according to professions and hierarchies with a defined scope and limitations in their functions. Abuse disrupts the functionality of the structure and, when not addressed despite explicit complaints, generates impunity, lack of accountability and consequences that harm both individuals and the organization. Tolerance to abuse tends to normalize behaviors and reproduce violence. The weak connection between the institutional and interpersonal levels fosters abusive relationships that are not conducive to work and learning.

When processes are disrupted in daily interactions, conflict emerges, and the environment becomes tense. In restrictive, exclusionary, disjointed and even immoral settings, the configuration of professional, personal and group identities is affected, and agents act defensively or are inhibited in their workplace and academic performance. As may be observed in the lower part of the diagram, abuse can be overt or covert. In the first case, aggressions are explicit, expressed through punishments, offenses, threats, discrimination, physical harm or sexual assault, among others. In covert abuse, symbolic violence in the environment is interpreted as uncertain. Agents may feel they are under stress, in a state of alert to resist threats, afraid, dissatisfied, tired, or underperforming. This corresponds to a climate of risk, insecurity, reprimands, and in most cases to work overload.

The text boxes with numbers (3.1., 3.2., 3.3 and 3.4) in the diagram correspond to the prompts in Section 3 of the questionnaire on clinical settings. Prompt 3.1, in the lower right part of the diagram (climate, feelings), asked residents to qualify their perception of the environment in which they interact. Prompt 3.2 is located in the same area of the diagram and consists of a single question that also reflects social tolerance to violence in the clinical setting. The set of statements in Prompt 3.3 appears at the top and bottom of the diagram, as these consider the relation between the institutional and interpersonal levels in the face of the phenomenon of abuse, especially in its explicit expressions. Lastly, Prompt 3.4 includes statements seeking to explain the connotations and meanings attributed to frequent workplace and educational practices in medical residences that may be perceived as violent.

#### About Perception and the Formulation of the Questions in Section 3

The epistemological assumption in the formulation of the questions in Section 3 was based on the concept of perception, understood as an active-constructive process in which those perceiving, before processing the new information, and with the data stored in their consciousness, build an anticipatory informational structure that allows them to contrast the stimulus—in this case the prompt—and to accept or reject it depending on whether it fits that structure [33]. Perception is the mental image formed with experiences and needs resulting from learning, selecting, organizing and interpreting stimuli and sensation processes. Individual perception is subjective, selective and temporary: subjective because the reactions to the same stimulus vary from one individual to another, selective because agents choose the perceptual field according to their motivations and interests, and temporary because it is short-term.

When developing questions about perception or opinion that imply a subjective value judgment, the selected response options corresponding to the affirmations expressed reflect the acceptance of a belief that may be considered valid on the basis of experience or evidence of certainties and that to a great extent depends on the context, i.e., on the knowledge of the reality under investigation. Perception questions include attitude-based evaluative components and belief-based cognitive components. However, attitudes are not always stable or detached from context and time. Individuals construct their opinions spontaneously based on different considerations that emerge at that moment rather than preconceived dispositions [34].

As Rasinski [35] explains, the expressions of attitudes are constructs that combine the contextual situation of the moment and the experiences deposited in the memory; the stable element lies in the evaluation. The effect of context operates in four phases: (1) comprehension, which refers to the understanding of the idea; (2) memory of previous experiences or opinions that are relatively stable and accessible when recalled; (3) judgment according to the inclusion/exclusion model [36] to apply evaluative standards; and (4) the report, consisting of questions grouped with measurement scales.

The RMLV questionnaire involved strong attitudes since experiences and knowledge about clinical settings are highly accessible. The relevance of the contextual information influences the construction of the evaluation. Depending on the responses, it is possible to understand the operative assessment: validation of the content of the statement with agreement, contradiction when there is disagreement or confusion when the information is conflicting and there is no defined attitude. In this regard, when preparing the statements and hypothetical situations, care was taken to ensure the clarity of the instructions, the presentation of the self-administered instrument, the phrasing of the statements, the order of the questions, specific situations in clinical contexts, previous knowledge of the target population, scales in the response options, types of technological devices for its application and possible attitudes of the residents toward the questionnaire.

In the section related to this study, Prompts 3.1 and 3.4 referred to the description of the reality of the clinical environment in which expressions of violence emerge, while Prompts 3.2 and 3.3 proposed hypothetical situations of violence that allowed us to explore possibilities and apply propositions. Because it was not our intent to implement a solution or resolve real-life situations, the responses could be more candid. This approach also made it possible for us to focus on a single dimension to assess its importance in context. Hypothetical situational affirmations allow respondents to establish priorities, desires and values about fictitious yet probable scenarios. In what follows, we present the results of the four prompts described with their corresponding items, investigating violence in clinical settings and their relationship with other variables collected in the questionnaire, such as gender, medical unit size by number of residents, specialty group and year of residency.

## 3. Results

The instrument had an 85% response rate, with 12,612 residents from 113 medical units or hospitals and 78 specialties participating. The data obtained from the RMLV questionnaire indicated the following reported prevalences of violence: 44.4% psychological, 32.7% academic-workplace, 6% sexual, 4.7% physical, 4.5% gender-identity-related, 3.4% cyber and 0.9% correlated with gender orientation. The reliability and internal consistency of violence in medical residencies were assessed with alpha omega, obtaining a value of 0.835 (CI; 0.828–0.843). Gender identity, specialty group, medical unit and year of residency (Table 1) stratified the sample. 

Upon analyzing the differences in PUEM resident perceptions regarding the characteristics of the environment by means of qualifying adjectives, we sought to describe the expansive–restrictive continuum of the clinical setting by specialty group (Table 2). Unpleasant, competitive, tense and conflictive environments were situated on the pole of restrictive settings, while pleasant, cooperative, relaxed and constructive environments tended toward expansive milieus. Neutral answers reflected undefined attitudes, which could indicate conflicting mindsets. Reading the data by specialty group, most residents in Internal Medicine described an expansive environment, although it is worth noting that when asked about the environment in terms of the tense–relaxed descriptors, the “neutral” value increased to 31.2%, and in the case of conflictive–constructive, 30.8% described it as “conflictive”. This pattern is repeated, with minor variations, in the Surgery, Gynecology, Pediatrics and Non-Clinical groups. Across all groups, between 60% and 70% considered the clinical environment pleasant, between 50% and 57% perceived it as cooperative, between 34% and 40% as relaxed and between 41% and 49% as constructive. In all cases, the lowest expansive figures were in Surgery and Gynecology. Therefore, these groups may be more prone to expressions of violence.

To determine whether the violence observed influences the perception of the clinical environment, one point was assigned to each of the five response options for the adjectives in Prompt 3.1 (unpleasant/pleasant, competitive/cooperative, tense/relaxed and conflictive/constructive), obtaining an average score of 9.65 +/− 4.11 with a range of 0.0 to 16.0. When comparing this index between residents who observed violence and those who had not observed it, we found that the latter scored higher in their perception of the clinical environment. This applies to all types of violence: psychological (10.80 vs. 8.55), physical (9.92 vs. 7.49), sexual (9.96 vs. 7.83) and academic-workplace (10.54 vs. 8.32), as may be observed in Table 3.

When relating gender identity and sexual orientation to the attitude toward discriminatory comments by others in their environment, the contrast between heterosexuals and those who are not was remarkable (Table 4). Although the numbers were relatively small, the percentages decreased by more than 25 points for the response option “most would disapprove”. In contrast, when considering the data according to sexual orientation, the difference between heterosexuals and other categories in the option “most would tolerate it” was at least four points. The perception of women regarding tolerance of discriminatory comments was also notable: 30.3% believed that such comments are tolerated in their clinical settings, while 27.2% of men perceived it as such.

Fear was one of the perceptions related to violence and the risk of experiencing it. In our analysis of responses to hypothetical situations of abuse by size of the medical unit, figures for all statements varied slightly at each end of the scale: in hospitals with up to 100 residents and in those with more than 500, compared to those with between 101 and 200 residents and 201 to 500 residents, respectively. Approximately 30% of the residents perceived risks involved in reporting violent events. This is related to the fact that approximately 50% believed that the aggressor would not be punished, which increased the sensation of impunity and vulnerability.

Other expressions that disrupt solidarity with the victims are reprisals against those who support them, which impedes reporting and promotes tolerance of violence in clinical settings. Nearly 30% believed that helping victims could have consequences for them. These same figures are repeated when inquiring about the institutional tolerance of abuse, which discourages reporting. Between 21% and 24% perceived that abuse was normal at their hospital, leading to constant fear in social relationships. In summary, in approximately one-third of the medical units, the clinical environments are intimidating and restrictive due to abuse and the different types of unpunished violence that can occur there. The graphic showing the perception of risk in hypothetical situations of violence in hospitals by number of residents and specialty may be consulted in section II of the Appendix A.

With respect to the set of question in prompt 3.4, we presented statements for residents to approve or reject based on their experience in clinical settings (see the figures in part III of the Appendix A). They were considered valid if the respondents agreed to some extent. For each statement, they were required to react, take a position and form an opinion that is assumed as evidence of certain characteristics of the learning environment and expressions of violence in the clinical context. Contrary to our expectations, the most frequent response option selected in all seven statements was “strongly disagree”. In what follows, a preliminary reading of data is presented horizontally and vertically. Observing the rows, in the first statement about punitive rotations to maintain order, the figures for the option “strongly disagree” exceeded 65%. It is important to note that in the columns, these numbers decrease as the years of residency progress, with a 29.5-point difference between R1 and R5, illustrating the disparity in the perception of actions, such as punitive rotations, that can become abusive. The figures related to recriminations for errors are more varied and decrease from R1 to R4 in the option “strongly disagree”, with only 40.6% of R5 stating they “somewhat agree” with the statement.

Regarding public questioning by professors, the responses ranged from 46.1% to 44.7% in R1 to R4 for “strongly disagree” and dropped to 34.4% in R5. As for performing extracurricular tasks, between 40% and 50% of the residents did not agree that it was better for them to do them to avoid conflict. More than half of the respondents did not agree that profanity and vulgar expressions exclude women from social dynamics. What most drew our attention was the figure of nearly 20% in the option “neither agree nor disagree”, reflecting confusion in the residents’ assessment of this statement. In terms of the affirmation alluding to the power of negative criticism among the community of peers that could cause the resignation of the discredited resident, nearly 30% stated that they “strongly disagree”; however, if we combine the options “somewhat agree” and “strongly agree” these figures (between 40% and 50%) exceed those in strong disagreement, indicating the relevance of violent social interactions in the clinical setting. Finally, for the question about the relationship between the strict application of the rules and a positive academic and workplace environment, between 34.4% and 24.4% responded that they did not agree these were related. These figures decreased as the year of residency progressed. As in other statements, the opinion of the R5 was more moderate and tended toward agreement.

## 4. Discussion

The central concepts guiding this study were learning environments and abuse in medical residencies. According to a review of the literature on violence in Latin American medical schools conducted by Mejía and Suárez [37], who retrieved publications from some 20 authors measuring expressions of violence, the prevalence of inappropriate behaviors ranged from 27% to 100%, the wide margin due to differences in the methodologies used. In our study, a prevalence of 52.3% was found in the PUEM medical residencies, with psychological (44.4%) violence being the most frequent manifestation.

Considering expressions of violence as social constructions [38], the modalities of social interactions between subjects in dynamic contexts are more varied. As Fuller and Unwin [9] explain, learning environments can be perceived on a continuum ranging between expansive and restrictive. This is based on the premise that environments where violence emerges inhibit learning, reduce educational potential and hold back the professional and personal development of physicians in training [5].

The contribution of this study is that it sought to describe the characteristics of clinical learning environments through the formulation of prompts based on a social, rather than individualistic, perspective. The strategy consisted of using qualifying adjectives and affirmations with hypothetical situations including expressions of explicit and covert forms of abuse that required residents to make value judgments and assess the veracity of the proposed statements. Because this way of characterizing the clinical environment has not been reported previously in the literature, our results cannot be compared with other studies conducted with the same methodology. Most publications on clinical learning environments worldwide, such as the work of Hernández Pérez and Bustillos Hernández [39], among others, are based on the application of the PHEEM scale (Postgraduate Hospital Educational Environment Measure) [40], and aimed at classifying the contextual conditions for learning, while the prompts presented here refer specifically to violence in clinical settings.

The perception of the organizational climate by specialty group on a continuum of antonymous adjectives to identify the positive and negative tendencies revealed that over 50% of residents in the Internal Medicine, Pediatrics and Non-Clinical specialty groups considered the environment expansive, while respondents from Surgery and Gynecology were a few points below the mean. These figures are of concern since approximately 30% answered with the neutral option, and 20% chose the negative adjective. The closest comparative figures are those of the study by Hamui et al. [21], which reported that the most expansive environments by specialty were found in Internal Medicine, followed by Family Medicine, Surgery, Gynecology-Obstetrics and lastly, Pediatrics. However, the measurement scale in this study was different, as it was focused not on evaluating violence but on learning.

Our study also found that women and gender minorities perceived greater tolerance for discriminatory comments in the clinical setting. However, we observed a more marked difference between heterosexuals and homosexuals, with the latter perceiving less frequency of disapproval of discriminatory comments in the clinical setting and greater tolerance and approval of these exclusionary expressions.

Patriarchal cultural representations of gender and inequality in power relations in medical training, as well as competitive interactions between residents, can foster environments where abuse and humiliation are incorporated into practices, which, as Peres et al. [39] explain, are reproduced and may even increase over time. Institutional level responses to the manifestations of violence on the interpersonal level that characterize clinical settings have been considered relevant [41] given that, as Bandura [42] observes, setting penalties for offenders limits violent practices and abuses of power. Through the posing of hypothetical situations in this study, we explored whether medical unit size by number of residents influenced the institutional tolerance of abuse. The perception of a lack of response from authorities was found to be greater in hospitals with fewer residents and those with more than 500 than in the two intermediate categories.

In the face of inaction, the risk of reporting abuse or helping victims increases, which inhibits individuals from lodging a complaint. One-third of the residents considered that abuse, which is related to restrictive environments for the professional development of medical specialists in training, was normal at their medical unit. In the bureaucratic and functional administration of hospitals, expressions of violence and abuse lead perpetrators to believe that they possess, in relation to the victim, a power derived from the position they occupy in the workplace [43]. Vertical hierarchies in the clinic put those at the bottom of the subordinate pyramid at risk of abuse. As Becher [44] notes, the predetermined “ethos” [45] in the educational environments of medicine, where the hidden curriculum supports an individualistic culture centered on study and suffering, creates an environment that influences the expression of violent behavior, tacitly but harmfully.

In addition, when relating hypothetical statements about common explicit and covert, formal and informal disciplinary measures in medical residencies to the year of residency, we observed that residents in their early years expressed disagreement with actions such as punitive rotations, public questionings, recriminations, extracurricular tasks, use of profanity and vulgar expressions, negative criticism from their peers or the strict application of the rules more frequently. As the years of residency progress, this perception tends to decrease. This is consistent with results from other studies; for example, Sheehan et al. [46] found that students who were frequently abused were less likely to complete assignments or provide optimal patient care than students who were not harassed.

Among the limitations of this study were the following: the results of the prompt on hypothetical situations demonstrate that the residents found little relation between the statements and what occurs in their clinical contexts, which will make it necessary to rework these to bring them closer to the reality of the environment under investigation. Possibly the interpretation from the perspective of violence studies on common practices seemed strange to them and they resisted understanding their experiences in this way. Another limitation was that, despite the large sample size, the high number of specialties, diverse health institutions and PUEM medical units, the survey was not administered nationwide; other educational entities would need to be included to complete the results. Finally, to explain some of the data presented above in-depth, such as the influence of hospital size on expressions of violence, it would be necessary to conduct qualitative research in specific contexts and even design intervention strategies to reduce expressions of violence in clinical environments.

## 5. Conclusions

In response to the research question guiding this study regarding how expressions of violence in clinical contexts alter learning environments, influencing the professional training of residents, we found that unpleasant, competitive, tense and conflictive contexts were related with threatening, unsafe and risky environments, and that this was more acute in specialty groups such as Surgery and Gynecology.

Returning to the five basic premises on violence [10] mentioned in the introduction, we consider that expressions of abuse are imbalances in the social order that need to be addressed. As demonstrated by the hypothetical situations described above, some forms of violence perform social functions of differentiation and integration and are considered part of the activities performed. Violence creates relational processes of (dis)articulation of meaning and practice and is socially defined in accordance with the criteria of regulations and morality in the clinical environment. These behavioral codes, that residents sometimes neither question, criticize or resist, are assumed or rejected in order to survive violence. In the face of abusive interactions, conventions are put into question, and conflicts eventually emerge.

In this study, we found that sexual orientation influences the perception of intolerance in the clinical setting with respect to discriminatory comments, such that for gender minorities, the environment was perceived as even more exclusionary. In terms of institutional responses, impunity for acts of violence generates insecurity, fear and the normalization of violence and leads to inaction, especially in hospitals with between 100 and 500 residents, constituting yet another element that contributes to promoting restrictive environments. Lastly, it may be affirmed that first-year residents perceive environments as more aggressive, a perception that tends to decrease in later years of residency.

The findings of this study have implications for research on violence studies and for the daily practices of residents in clinical settings. In the first case, the methodological contribution consisted of the conceptual framework that situates the problem with a social and not only an individual perspective, highlighting the institutional and interactional dimension. It also proposes new items to describe the characteristics of clinical learning environments, for example, the use of adjectives to represent the workplace climate, the measurement of tolerance to violence and the perception of abuse situations are tools that allow us to approach the clinical environment. On the other hand, from the point of view of practice, the results warn about aspects that should be institutionally addressed, including abuses in hierarchical systems, surveillance at specific times and places where interaction with residents are intensified, as well as being aware of the perspective of women and gender minorities.

Abuses in power relations, rigid hierarchical positions, offensive clinical interactions and inappropriate language in conversations may foster restrictive environments. In such settings, interactions loaded with socio-culturally learned violence can be reproduced; however, as social constructions, asymmetrical relationships resulting in inappropriate behaviors may be deconstructed and transformed through diverse and simultaneous strategies. Such strategies could include the non-violent communication proposed by Rosenberg [47], modeling peaceful behaviors [48], the application of effective institutional sanctions [4], collective reflections on violent events, guided support groups for victims that could foster new relationships, the constant observation of bystanders located in the clinic who testify to compromising situations [49], the repair of damages and the mediation between the parties in conflict [50], among others. All these strategies are focused on making violence visible, promoting dialogue and reflection and are aimed at reducing impunity. Ultimately, what is sought is to restore the social fabric in restrictive clinical contexts and to build expansive, secure and inclusive clinical climates with medical residencies free of violence.

## Figures and Tables

**Figure 1 ijerph-20-06754-f001:**
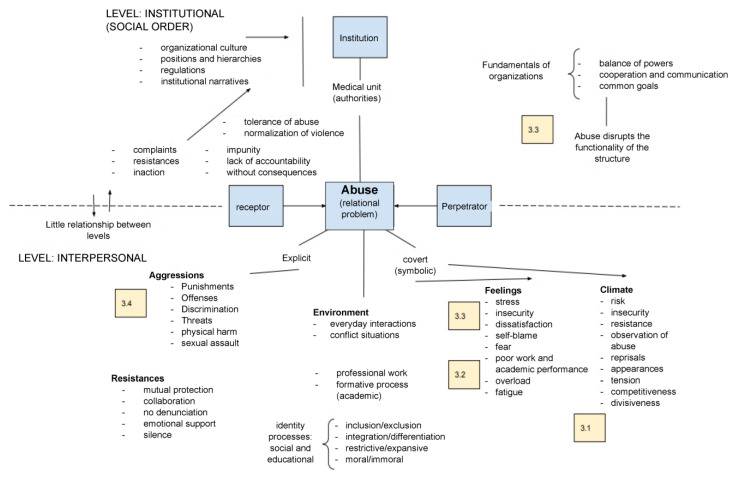
Conceptual framework of clinical environments and abuse. Source: designed by the authors.

**Table 1 ijerph-20-06754-t001:** Demographic characteristics of UNAM resident physicians.

Demographic Characteristic	Total Sample *n* (%)
Gender Identity	
Male	6201 (49.2)
Female	6261 (49.6)
LGBTI+ *	73 (0.6)
Prefer not to answer	79 (0.6)
Specialty Group	
Internal Medicine	4539 (36.0)
Surgery	3933 (31.2)
Gynecology	1109 (8.8)
Pediatrics	2033 (16.1)
Non-Clinical	998 (7.9)
Year of Residency	
1st	5148 (40.8)
2nd	3664 (29.1)
3rd	2517 (20.0)
4th/5th	1264 (10.0)
No response	19 (0.1)

* LGBTI+: Abbreviation for lesbian, gay, bisexual, transgender and intersex+. Source: by the authors.

**Table 2 ijerph-20-06754-t002:** Perception of the environment by specialties.

		Specialty										
		Internal Medicine		Surgery		Gynecology		Pediatrics		Non-Clinical		
		N	%	*n*	%	*n*	%	N	%	*n*	%	*X*^2^ (*p*)
Unpleasant–Pleasant Environment	Unpleasant	755	**16.6**	640	16.3	177	16.0	271	13.3	155	15.5	90.516 (0.001)
	Neutral	750	16.5	886	22.5	262	**23.6**	341	16.8	182	18.2	
	Pleasant	3034	66.8	2407	61.2	670	60.4	1421	**69.9**	661	66.2	
Competitive–Cooperative Environment	Competitive	878	19.3	768	19.5	221	**19.9**	369	18.2	195	19.5	28.38 (0.001)
	Neutral	1199	26.4	1179	**30.0**	329	29.7	559	27.5	240	24.0	
	Cooperative	2462	54.2	1986	50.5	559	50.4	1105	54.4	563	**56.4**	
Tense–Relaxed Environment	Tense	1325	29.2	1211	30.8	356	**32.1**	596	29.3	271	27.2	20.248 (0.009)
	Neutral	1414	31.2	1297	33.0	367	33.1	677	**33.3**	329	33.0	
	Relaxed	1800	39.7	1425	36.2	386	34.8	760	37.4	398	**39.9**	
Conflictive–Constructive Environment	Conflictive	1397	30.8	1226	**31.2**	337	30.4	584	28.7	275	27.6	48.253 (0.001)
	Neutral	1053	23.2	1091	27.7	308	**27.8**	500	24.6	234	23.4	
	Constructive	2089	46.0	1616	41.1	464	41.8	949	46.7	489	**49.0**	

Note: significant differences (*p* < 0.05) were obtained with the two-sided equality test for column proportions. Bonferroni correction was used. Differences are shown with percentages highlighted in bold. The tests assume equal variances. Source: by the authors.

**Table 3 ijerph-20-06754-t003:** Influence of violence observed in the clinical environment.

			Environment of the Clinical Site	
Violence		*n*	M (SD)	t (*p*)
Psychological	No	6179	10.80 (4.01)	31.84 (0.001)
	Yes	6433	8.55 (3.91)	
Physical	No	11,227	9.92 (4.04)	20.83 (0.001)
	Yes	1385	7.49 (4.11)	
Sexual	No	10,790	9.96 (4.06)	20.99 (0.001)
	Yes	1822	7.83 (4.00)	
Academic-Workplace	No	7561	10.54 (4.03)	31.08 (0.001)
	Yes	5051	8.32 (3.87)	

Source: by the authors.

**Table 4 ijerph-20-06754-t004:** Tolerance of the environment for discriminatory comments. What do you think would be the most typical reaction in your direct and daily environment on site (peers, colleagues) to a discriminatory comment? (Mark one option only).

	Most Would Disapprove		Most Would Tolerate It		Most Would Approve and Continue with the Comment		
	*n*	row %	*n*	row %	*n*	row %	*X^2^* (*p*)
Gender Identity							
Male	3542	57.1	1687	27.2	972	15.7	58.86 (0.001)
Female	3513	56.1	1894	30.3	854	13.6	
Transgender Male/Trans Male/Female to Male (FTM)	1	14.3	5	71.4	1	14.3	
Transgender Woman/Trans Woman/Male to Female (MTF)	0 ^1^	0.0	4	100.0	0 ^1^	0.0	
Genderqueer, neither exclusively male nor female, non-binary	17	33.3	24	47.1	10	19.6	
Additional gender category/(or other)	3	33.3	4	44.4	2	22.2	
Prefer not to answer	33	41.8	26	32.9	20	25.3	
Sexual Orientation							
Heterosexual (not gay, not lesbian)	6461	57.4	3162	28.1	1626	14.5	63.44 (0.001)
Lesbian, gay or homosexual	383	51.1	259	34.5	108	14.4	
Bisexual	136	42.2	124	38.5	62	19.3	
Something else	14	45.2	11	35.5	6	19.4	
Don’t know	14	48.3	9	31.0	6	20.7	
Prefer not to answer	101	43.7	79	34.2	51	22.1	

Note: ^1^ This category was not used in comparisons because its column proportion equals zero or one. Significant differences (*p* < 0.05) were obtained with the two-sided equality test for column proportions. Bonferroni correction was used. The tests assume equal variances. Source: by the authors.

## Data Availability

The data presented in this study are available on request from the first author.

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
