# Peer review of "Violence and Clinical Learning Environments in Medical Residencies"

_ijerph, 2023, doi:10.3390/ijerph20186754_

Round 1

Reviewer 1 Report

This study aims to investigate the impact of violence and abuse on the learning environment in hospital settings. It is an important topic, however, the manuscript needs to be revised

 To establish a stronger connection between violence/abuse and the learning environment in hospital settings, it is crucial to highlight the findings of previous research. How has prior literature addressed this relationship? What gaps remain in the existing literature and research that justify conducting this present study? Furthermore, to enhance accessibility, the authors most emphasize more English references in the literature review, as opposed to predominantly Spanish references.

The authors claim that different types of violence often emerge in restrictive environments in which power relations are more pronounced and communication less fluid. Furthermore, the authors claim that workers in the health sector are at risk of suffering acts of interpersonal violence in their workplaces. However, it is not demonstrated clearly enough if and how previous research has shed light on these associations – e.g between violence/abuse and the learning environment in hospitals and the risk of interpersonal violence at hospitals. The introduction must better explain why there is still room for comprehensive investigations that can address certain gaps. Although several Spanish references may contribute to this topic, including English references to broaden the readership of this study is needed.

 Method:

 This study is based on quantitative data collected from a separate research endeavor. However, it is essential to provide more detailed information on the execution of that study, including when, how, and where it was conducted. While some aspects of the previous research were briefly discussed in the discussion section, this were not adequately presented earlier.

 Participants: Clearly outline the characteristics of the participants, such as medical professionals, students, or patients, and provide details on how they were recruited.

 Statistical Analyses: Elaborate on the statistical methods employed to analyze the data and the rationale behind their selection. For instance, why are t-tests and Chi-tests used instead of regression analyses, so relevant confounders can be adjusted for?  Please explain: The reliability and internal consistency of the instrument were assessed by "alpha omega" – what is an alpha omega test?

 The study is a cross-sectional study, so the study can only report correlations and not associations the study is a cross-sectional study, so the study can only report correlations and not associations. It must be clarified in more in the discussion. In the abstract the authors conclude: Abuses in power relations, rigid hierarchical positions, offensive clinical interac-28 tions and inappropriate language in conversations foster restrictive environments. In such settings, 29 interactions loaded with socio-culturally learned violence are reproduced; however, as social con-30 structions, asymmetrical relationships resulting in inappropriate behaviors may be deconstructed 31 and transformed through diverse and simultaneous strategies. Greater caution must be exercised with the conclusions of the study due to the cross-sectional design and the analyses. 

 Figure 1 - Study Framework: Considering the model's complexity and potential confusion for readers, it might be best to omit this section.

 Table 1 and Table 4: Clarify the meaning of "a," "b," and "c" in the tables to enhance reader comprehension.

 Table 5: Translate the contents of the table from Spanish into English to ensure accessibility to a broader audience.

Finally, to enhance the overall quality of the study, it is necessary to increase the number of English references and ensure that the majority of the sources are readily available to readers. Many Spanish references are used.

Author Response

Reviewer 1

Comments and Suggestions for Authors

This study aims to investigate the impact of violence and abuse on the learning environment in hospital settings. It is an important topic, however, the manuscript needs to be revised.

 To establish a stronger connection between violence/abuse and the learning environment in hospital settings, it is crucial to highlight the findings of previous research. How has prior literature addressed this relationship? What gaps remain in the existing literature and research that justify conducting this present study? Furthermore, to enhance accessibility, the authors most emphasize more English references in the literature review, as opposed to predominantly Spanish references.

We added 19 articles in English related to violence and clinical learning environments. They are cited throughout the text and in the final bibliography.

The authors claim that different types of violence often emerge in restrictive environments in which power relations are more pronounced and communication less fluid. Furthermore, the authors claim that workers in the health sector are at risk of suffering acts of interpersonal violence in their workplaces. However, it is not demonstrated clearly enough if and how previous research has shed light on these associations – e.g between violence/abuse and the learning environment in hospitals and the risk of interpersonal violence at hospitals. The introduction must better explain why there is still room for comprehensive investigations that can address certain gaps. Although several Spanish references may contribute to this topic, including English references to broaden the readership of this study is needed.

We included English references (17 to 25) at the introduction that shed light on the relation between violence and restrictive environments. This study aims to delve into the characteristics of residents in learning environments by gender, specialty, year of residence and size of the medical unit by number of residents.

 Method:

 This study is based on quantitative data collected from a separate research endeavor. However, it is essential to provide more detailed information on the execution of that study, including when, how, and where it was conducted. While some aspects of the previous research were briefly discussed in the discussion section, this were not adequately presented earlier.

The following text was added on lines 161-170

     “The research was planned and implemented from the Postgraduate Studies Division (DEP) at the National Autonomous University of Mexico (UNAM) Faculty of Medicine. From June to December 2021, the literature was reviewed and a self-administered questionnaire was designed taking into account different violence scales such as the study of sexual and gender harassment conducted at the Complutense University of Madrid [31], as well as the structure and operation of medical residences in Mexican health institutions. The questionnaire was reviewed and tested with social service students to validate its content and understanding of the questions, and adjustments were made accordingly. The instrument was digitally programmed at the Computer Department.”

 Participants: Clearly outline the characteristics of the participants, such as medical professionals, students, or patients, and provide details on how they were recruited.

The following text was rewritten on lines 171 – 182

   “Participants in the study were the registered residents in the 2021-2022 cycle who presented the annual exam of the Academic Program for Medical Specialties (Plan Único de Especializaciones Médicas, PUEM) of (UNAM) Faculty of Medicine. When they accessed the website to consult their evaluations they were asked to complete the survey. The inclusion criteria considered the 12,900 registered residents at PUEM, so it was more of a census. 6,827 were women (52.92%) and 6,073 men (47.08%), distributed in 78 medical specialties, in 156 medical units and 686 programs. Residents from other universities academic programs were excluded. The questionnaire was available on January 2022. They answered the survey voluntarily after approving the informed consent at the beginning of the instrument. The responses were received in real time in the digital database platform. At the end of the reception period, data was reviewed to correct errors. Once clear statistical analyzes were carried out according to the objectives of the study.”

 Statistical Analyses: Elaborate on the statistical methods employed to analyze the data and the rationale behind their selection. For instance, why are t-tests and Chi-tests used instead of regression analyses, so relevant confounders can be adjusted for?  Please explain: The reliability and internal consistency of the instrument were assessed by "alpha omega" – what is an alpha omega test?

The statistical analyses section was rewritten as appear in lines 236 -255

    “The statistical package SPSS v.23.0 (IBM Corp. Armonk, NY) was used to analyze the data obtained. Descriptive statistics was used in the analysis and bivariate comparisons were made with Chi-squared and t-tests. Contingency tables have a moderating effect on the variables of interest, such as the type of violence (sexual, physical, psychological, academic, labor and gender), medical specialty and perception of risk of violence, and their relation with clinical learning environments. For this work, we did not seek to develop an explanatory model, due to the possible number of confounding variables to be taken into account in the study variable (clinical setting) and their impact on the accuracy of the model. Comparisons between proportions were performed using the z-test and Bonferroni correction. A = a.05 was considered statistically significant.

    For this reason, using contingency tables, the moderating effect that variables of interest, such as the type of violence (sexual, physical, psychological, academic, labor and gender), the medical specialty and the perception of risk of violence, could be verified.

    For this article, we considered Section 3 of the RMLV questionnaire to describe violence in clinical learning environments and to situate them on the expansive-restrictive continuum. The reliability and internal consistency of violence in medical residencies were assessed by alpha omega, an internal consistency estimator, more sensitive than other estimators (like Coeficient α, β, H Ordinal) based on factor loadings that indicate the proportion of the variance attributed to the common variance and does not overestimate reliability [32]. The value obtained was = 0.835 (CI; 0.828‒0.843).”

 The study is a cross-sectional study, so the study can only report correlations and not associations the study is a cross-sectional study, so the study can only report correlations and not associations. It must be clarified in more in the discussion. In the abstract the authors conclude: Abuses in power relations, rigid hierarchical positions, offensive clinical interactions and inappropriate language in conversations foster restrictive environments. In such settings, interactions loaded with socio-culturally learned violence are reproduced; however, as social constructions, asymmetrical relationships resulting in inappropriate behaviors may be deconstructed and transformed through diverse and simultaneous strategies. Greater caution must be exercised with the conclusions of the study due to the cross-sectional design and the analyses. 

We changed the word “associated” for “related” in all the article.

We refrased the cited sentences (in the abstract lines 58-63, in conclusions lines 66-621) as follow:

     “Abuses in power relations, rigid hierarchical positions, offensive clinical interactions, and inappropriate language in conversations may foster restrictive environments. In such settings, interactions loaded with socio-culturally learned violence can be reproduced; however, as social constructions, asymmetrical relationships resulting in inappropriate behaviors may be deconstructed and transformed through diverse and simultaneous strategies.”

 Figure 1 - Study Framework: Considering the model's complexity and potential confusion for readers, it might be best to omit this section.

We consider that the conceptual framework presented in the diagram and the explanation that follows (lines 276 – 313) are a contribution to the study of violence in clinical environments, so we decided not to omit it.

 Table 1 and Table 4: Clarify the meaning of "a," "b," and "c" in the tables to enhance reader comprehension.

The subscripts a, b, c were removed from tables 2, 4 and 5.

 Table 5: Translate the contents of the table from Spanish into English to ensure accessibility to a broader audience.

The words in Figure 2 were translated to English.

Finally, to enhance the overall quality of the study, it is necessary to increase the number of English references and ensure that the majority of the sources are readily available to readers. Many Spanish references are used.

We added 19 articles in English related to violence and clinical learning environments. They are cited throughout the text and in the final bibliography.

Reviewer 2 Report

The study clearly defines a conceptual model for the analysis of violence in the context of a hospital clinic, which makes it possible to objectively interpret the relationship between the abuser and the abused, as well as the influence of the environment on both. It presents an original method to approach the subject, generating important results to understand the phenomenon and infer possible solutions to the problem.

Suggestion: Figure 2 should be revised in order to maintain coherence in the language used in the writing of the article.

Author Response

Suggestion: Figure 2 should be revised in order to maintain coherence in the language used in the writing of the article.

The words in Figure 2 were translated to English.

Reviewer 3 Report

The manuscript is intended to provide a description and analysis of the residents’ perceptions of characteristics on the expansive/restrictive continuum of the clinical learning environment where they work and study.

The paper is well-written and articulated, in particular the Methodology section.

However, I suggest some general comments in what follows:

I believe that one of the main flaws of the manuscript is that lacks adequate reference to the literature (examples are lines 38-41, lines 48-49, lines 61-64, etc.). The entire section after Figure 1 (lines 195-243), although interesting, also lacks reference to the literature. The final references are not so many, and this might compromise the strength of the epistemological foundations on which the study is based.

The methodology is well explained and the sample size is very big, which adds to the scientific soundness of the manuscript.

Lines 310-311: how may these characteristics (i.e., “the precision and pace of the surgical skills required”) of Surgery and Gynecology be associated with increased violence? (ps. Be aware of the typo in this expression - "thee")

The Figure on p. 11 should be translated into English.

Lines 380-385: this interesting finding should be accompanied by an adequate interpretation in the Discussion section.

Finally, I believe that a section on the Implications for research and practice might improve the manuscript.

Author Response

I believe that one of the main flaws of the manuscript is that lacks adequate reference to the literature (examples are lines 38-41, lines 48-49, lines 61-64, etc.). The entire section after Figure 1 (lines 195-243), although interesting, also lacks reference to the literature. The final references are not so many, and this might compromise the strength of the epistemological foundations on which the study is based.

On the first comment, we added 19 articles in English related to violence and clinical learning environments. They are cited throughout the text, including the referred lines, and in the final bibliography.

Figure 1 and the text that follows (195 – 243) are a contribution of the authors, that is why there are no references, the text is the explanation of the diagram.

The methodology is well explained, and the sample size is very big, which adds to the scientific soundness of the manuscript.

Lines 310-311: how may these characteristics (i.e., “the precision and pace of the surgical skills required”) of Surgery and Gynecology be associated with increased violence? (ps. Be aware of the typo in this expression - "thee")

The sentence “the precision and pace of the surgical skills required” was omitted.

The Figure on p. 11 should be translated into English.

The words in Figure 2 were translated to English.

Lines 380-385: this interesting finding should be accompanied by an adequate interpretation in the Discussion section.

Lines 380 – 385 refer to the perception of violence by year of residence, on the discussion there are arguments (lines 562 – 566) about the effects of violence on residents on their first years. I would appreciate some clues about the meaning of “adequate interpretation”.

Finally, I believe that a section on the Implications for research and practice might improve the manuscript.

In the conclusion section a paragraph (lines 604 -615) was added to emphasize the implications for research and practice. The text reads as follo:

   “The findings of this study have implications for research on violence studies and for the daily practices of residents in clinical settings. In the first case, the methodological contribution consisted of the conceptual framework that situates the problem with a social and not only an individual perspective, highlighting the institutional and interactional dimension. It also proposes new items to describe the characteristics of clinical learning environments, for example, the use of adjectives to represent the workplace climate, the measurement of tolerance to violence and the perception of abuse situations are tools that allow us to approach the clinical environment. On the other hand, from the point of view of practice, the results warn about aspects that should be institutionally addressed, including abuses in hierarchical systems, surveillance at specific times and places where interaction with residents are intensified, as well as being aware of the perspective of women and gender minorities.”

Reviewer 4 Report

Abstract:

 method part:

Please briefly explain the questionnaire.

"we considered Section 3 of the 21 instrument on clinical environments and violence" what were section 1 or two.??

How many questions did the instrument have and what dimensions did it include?

Page 2 introduction needs reference.

For example: The five basic premises....

Lines 67-70, 75-77

The specific objectives were as follows should be removed from the end of introduction and go to the method section

Transfer table one to the result section.

In the method section, please describe the tool in detail. What were the dimensions? How many questions did you have? What was the question scoring system? yes/no Or Likert scale?

What was the sampling method?

What was the inclusion criteria? What was the exclusion criteria?

Abbreviations should be explained. LGBTI+? UNAM??

The data obtained from the RMLV questionnaire indicated the following reported 143 prevalences of violence: 44.4% psychological, 32.7% academic-workplace, 6% sexual, 4.7% 144 physical, 4.5% gender-identity-related, 3.4% cyber and 0.9% associated with gender ori-145 entation. The reliability and internal consistency of violence in medical residencies were 146 assessed by alpha omega, obtaining a value of = 0.835 (CI; 0.828‒0.843)

Why is reported in the analysis section. In this section, explain how to analyze the data in the study.

For this article, we considered Section 3 of the RMLV questionnaire to describe vio-148 lence in clinical learning environments and to situate them on the expansive-restrictive 149 continuum. What was section one and two?

Section 3 considered four sets of questions on clinical environments

Move from lines 155 to 174 to the method section.

Tables 3.1 to 3.4 should be deleted. However, the method of scoring, the range of questions in each section, the number of questions in each section should be explained descriptively in the method section.

Reference figure 1? Designed by the authors?

Lines 195 to 294???

Figure 2 is not in English and cannot be understood.

Tables need to be redesigned. What does table number 5 indicate???

Author Response

Abstract:

Method part:

Please briefly explain the questionnaire.

The following text was added on the abstract

“The instrument was structured in eight sections: introduction and informed consent, gender identity, sexual orientation, clinical environment, experienced types of violence, gender discrimination and cyberagressions, effects of violence in the health of residents and sociodemografic data.”  

The following text was added on the Method part

“The structure of the RMLV questionnaire consisted of an introduction and eight sections. The introduction contained preloaded data obtained from the DEP databases: UNAM student account number (resident ID), year of residency, specialty and subspecialty, institution and medical unit. This section also included a presentation of the study objective, a checkbox confirming their voluntary participation and a hyperlink with the full informed consent document. First and second sections (one question each) explored gender identity (how they self-identify) and sexual orientation (to whom they are attracted) in multiple-choice questions. Third section (17 questions) referred to the environment at the clinical unit and its description through qualifying adjectives and hypothetical situations related to expressions of violence, abuse and harassment. Fourth (28 questions), fifth (four question) and sixth (five questions) dealt with types of violence, gender discrimination and cyber-aggression. These questions were structured in a matrix and constituted the core of the questionnaire. The possibility of having experienced violence was explored, and if affirmative, the frequency, severity, location where it took place, time of the event, the perpetrator of the abuse, the actions carried out, reporting mechanisms and institutional response were explored with multiple-choice questions. Seventh section (four questions) alluded to the effects of violence on the health of residents, on their state of mind, the support available to address their mental health and the impact of violence on continuing the residency. Finally, the eighth dimension (eight questions) collected sociodemographic data of residents. The scoring systems were diverse in each section depending on the question, sometimes yes or no, Likert scales and sentence selection were also used. The questionnaire in Spanish may be accessed at the following link: https://www.dropbox.com/s/8nqoc5sddzubvob/Cuestionario%20RMLV%20para%20protocolo%2028-01-2022.pdf?dl=0”

"we considered Section 3 of the 21 instrument on clinical environments and violence" what were section 1 or two.??

First and second sections (one question each) explored gender identity (how they self-identify) and sexual orientation (to whom they are attracted) in multiple-choice questions. It was specified on the questionnaire description.

How many questions did the instrument have and what dimensions did it include?

The instrument have 58 questions in eight sections as described in the added paragraph about the questionnaire.

Page 2 introduction needs reference. For example: The five basic premises.... Lines 67-70, 75-77

We included 19 articles in English related to violence and clinical learning environments. They are cited primary at the introduction, throw-out the article and in the final bibliography.

The specific objectives were as follows should be removed from the end of introduction and go to the method section

The specif objectives have been moved from the introduction to the beginning of the method section

Transfer table one to the result section.

Table one has been moved to the result section

In the method section, please describe the tool in detail. What were the dimensions? How many questions did you have? What was the question scoring system? yes/no Or Likert scale?

The instrument dimensions and scoring system has been described in the added paragraph about the questionnaire.

What was the sampling method?

There was not a calculated sample, the study was more like a census, since all the PUEM residents were included in the study.

What was the inclusion criteria? What was the exclusion criteria?

The following lines were added in the method section to answer the question:

“The inclusion criteria considered the 12,900 registered residents at PUEM, so it was more of a census. 6,827 were women (52.92%) and 6,073 men (47.08%), distributed in 78 medical specialties, in 156 medical units and 686 programs. Residents from other universities academic programs were excluded.”

Abbreviations should be explained. LGBTI+? UNAM?? 

We added a note in Table 1: *LGBTI+: Abbreviation for Lesbian, Gay, Bisexual, Transgender, and Intersex+ 

The data obtained from the RMLV questionnaire indicated the following reported prevalences of violence: 44.4% psychological, 32.7% academic-workplace, 6% sexual, 4.7% physical, 4.5% gender-identity-related, 3.4% cyber and 0.9% associated with gender orientation. The reliability and internal consistency of violence in medical residencies were assessed by alpha omega, obtaining a value of = 0.835 (CI; 0.828‒0.843). Why is reported in the analysis section. In this section, explain how to analyze the data in the study.

The referred data was moved from the method to the results section were the analyze strategies were described.

For this article, we considered Section 3 of the RMLV questionnaire to describe violence in clinical learning environments and to situate them on the expansive-restrictive continuum. What was section one and two?

First and second sections (one question each) explored gender identity (how they self-identify) and sexual orientation (to whom they are attracted) in multiple-choice questions. It was specified on the questionnaire description.

Section 3 considered four sets of questions on clinical environments. Move from lines 155 to 174 to the method section.

The prompts with section 3 questions has been moved from the introduction to the method section

Tables 3.1 to 3.4 should be deleted. However, the method of scoring, the range of questions in each section, the number of questions in each section should be explained descriptively in the method section.

The text box with Prompts 3.1 to 3.4 has been removed from the main texts and added as supplementary material

Reference figure 1? Designed by the authors?

Yes, designed by the authors, it has been explicitly written at the end of the Figure 1.

Lines 195 to 294???

The indicated lines correspond to the explanation of the elements presented on Figure 1 and their relationship with the clinical environment section 3 of the study. The 4 prompts are situated in the conceptual framework to understand the interactions on the hospital.

Figure 2 is not in English and cannot be understood.

The words in Figure 2 were translated to English

Tables need to be redesigned. What does table number 5 indicate???

Table 5 refers to the responses in Prompt 3.4 on the degree of agreement with the statements about common inappropriate practices in medical residencies.

Round 2

Reviewer 1 Report

The authors have done a fine job in addressing previous criticism. However, there are two points that could have been handled better:

In relation to an international journal, there are still quite a few Spanish references.

The introduction could be developed to make it clearer what this study can contribute. What is the current state of research with regard to the posed questions? What are the limitations of existing studies (and there may not be any studies that have examined this previously, so please state that), and what scientific gap this study aims to fill?

The abstract should be shortened.

Author Response

Thank you for your comments, here are the responses to the three indicated points:

In relation to an international journal, there are still quite a few Spanish references.

We removed references 38 to 56 and rearranged the citations and bibliography numbers.

The introduction could be developed to make it clearer what this study can contribute. What is the current state of research with regard to the posed questions? What are the limitations of existing studies (and there may not be any studies that have examined this previously, so please state that), and what scientific gap this study aims to fill?

A new text was written in blue letters inserted as the second paragraph of the introduction answering the questions posed by the reviewer.

The abstract should be shortened.

The abstract was reduced to 200 words, it appears in the manuscript in blue letters.

Reviewer 4 Report

Thanks to the authors for the revision

Figure No. 2 and Table No. 5 should be deleted.

The most important data should be included in the text.

Table 5 and figure 2 do not help the reader.

Author Response

Thank you for your comments. With respect to your suggestion:

"Figure No. 2 and Table No. 5 should be deleted.

The most important data should be included in the text.

Table 5 and figure 2 do not help the reader"

Figure 2 and Table 5 were removed from the main manuscript and added as supplementary material. We decided that the reader should have the possibility to consult the figures related to the explained data in the results and discussion sections. This was indicated in the text in blue letters.